# Generating Behaviorally Diverse Policies with Latent Diffusion Models

**Shashank Hegde**[*]
University of Southern California
khegde@usc.edu

**Sumeet Batra** [*]
University of Southern California
ssbatra@usc.edu

**K.R. Zentner**
University of Southern California
kzentner@usc.edu

**Gaurav S. Sukhatme**[†]
University of Southern California
gaurav@usc.edu

## Abstract

Recent progress in Quality Diversity Reinforcement Learning (QD-RL) has enabled learning a collection of behaviorally diverse, high performing policies. However, these methods typically involve storing thousands of policies, which results in high space-complexity and poor scaling to additional behaviors. Condensing the archive into a single model while retaining the performance and coverage of the original collection of policies has proved challenging. In this work, we propose using diffusion models to distill the archive into a single generative model over policy parameters. We show that our method achieves a compression ratio of 13x while recovering 98% of the original rewards and 89% of the original humanoid archive coverage. Further, the conditioning mechanism of diffusion models allows for flexibly selecting and sequencing behaviors, including using language. Project website: https://sites.google.com/view/policydiffusion/home.

## 1 Introduction

Quality Diversity (QD) is an emerging field in which collections of high performing, behaviorally diverse solutions are trained. QD methods perform what is often referred to as illumination or divergent search, in that they attempt to illuminate the search space rather than optimizing towards a single point. QD algorithms have shown success in learning robot controllers capable of adapting to damage, solving hard exploration problems, and generating diverse scenarios in the procedural content generation (PCG) domain [5] [7] [2]. The foundational method, Map Elites [21], maintains an archive of solutions where each cell in the archive corresponds to a solution with a score given by the task objective $f$, and behavior specified by measure functions $m_1...m_k$, which map to a low dimensional behavior space. The measure functions $m_1...m_k$ specify which cell each solution belongs to in the $k$ dimensional archive. New solutions are evolved using evolutionary methods and inserted into the archive only if they outperform existing ones with the same behavior.

A promising subclass of methods (Quality Diversity Reinforcement Learning (QD-RL)) combines the optimization capabilities of RL with the illumination capabilities of QD, to find high performing and diverse solutions. In the robotics domain where the environment and the objective-measure functions $f$ and $\mathbf{m}$ are assumed to be non-differentiable, RL can be leveraged to estimate the gradients of $f$ and/or $\mathbf{m}$ and provide a powerful performance improvement operator on the solutions in the

---

[*]Equal contribution
[†]Sukhatme holds concurrent appointments as a Professor at USC and as an Amazon Scholar. This paper describes work performed at USC and is not associated with Amazon.

37th Conference on Neural Information Processing Systems (NeurIPS 2023).

archive. QD-RL methods that combine QD with on-policy and off-policy RL algorithms have shown promising results on a variety of locomotion tasks and are capable of finding a plethora of high-quality gaits [22] [28] [23] [1]. One of several drawbacks of existing QD-RL methods is that they must maintain a collection of hundreds, if not thousands of policies, in order to cover the behavior space, which leads to poor space-complexity and difficulty in real-world deployment. Map-Elites-based QD methods show poor scaling properties and suffer from the curse of dimensionality, quite literally in that as the dimensionality $k$ of the archive increases, the number of solutions one needs to store increases exponentially. Prior methods have attempted to scale Map-Elites to higher dimensional problems by using Centroidal Voronoi Tessellations to divide the archive into a small number of evenly spaced geometric regions [31]. However, these methods require recomputing the Voronoi cells periodically, resulting in worse runtime performance, and try to keep the number of niches small in order to effectively exploit local competition. In order to smoothly interpolate between behaviors of different solutions with a discrete archive, one must upsample the archive resolution to (tens of) thousands of policies, often resulting in a level of discretization higher than the actual occurrence of distinct behaviors, while further worsening the space and time complexity of the algorithm.

An alluring idea is to distill the archive into a single, expressive model that completely covers the behavior space and maintains high performance. A single model representing the archive reduces space-complexity and potentially allows for smooth interpolation in the behavior space, making it easier to deploy and use in downstream tasks. Prior methods have shown promising results in distilling the archive into a single policy [8], or by learning a generative model via adversarial training over the policies in the archive [16]. We wish to improve on these methods by maintaining, or even improving, the overall performance of the original archive during the distillation phase, and scale generative models to be able to represent policies parameterized by deep neural networks rather than a low dimensional 1D vector of parameters.

To this end, we utilize the powerful generative and conditioning mechanisms of diffusion models to distill the archive into a single, expressive model that can generate a policy with any behavior from the behavior space mapped by the original archive. This generative process can be conditioned on the desired behavior measures, and even language descriptions of the desired behavior. Diffusion models have shown great success in computer vision in image quality and diversity [15] [6]. Latent diffusion models accelerate training by compressing the image dataset into a compact, expressive latent space and training a diffusion model on this lower dimensional space [25]. They proceed in two stages, first by compressing imperceptible, high-frequency details via a learned dimensionality reduction, and then by learning the semantic details of the images via the actual diffusion model itself. Similarly, here we show that one can compress a collection of policies parameterized by deep neural networks into a lower dimensional space by using a variational auto encoder (VAE), and then learn the semantic or behavioral details of the policy distribution via latent diffusion. Our experiments show evidence of the manifold hypothesis or the elite hypervolume [32], that all high performing policies lie on a low-dimensional manifold. We summarize our contributions below.

1. We compress an archive of policies parameterized by deep neural networks and trained via a state of the art QD-RL method PPGA into a single, expressive model while maintaining performance of the policies in the original dataset.
2. We use the iterative conditioning mechanism of diffusion models to reconstruct policies with precise locations in measure space, and demonstrate how language conditioning can be used to flexibly generate policies with different behaviors.
3. We showcase our model's ability to sequentially compose completely different behaviors together, and additionally show that language conditioning can be used to dramatically improve the performance and consistency of sequential behavior composition.

## 2 Related Work

**Quality Diversity**    QD Optimization attempts to illuminate the search space with high performing solutions. The optimization problem is formulated as follows. Given an objective $f(\cdot)$ to maximize and $k$ measure functions $\mathbf{m} = < m_1(\cdot)...m_k(\cdot) >$ that map a solution $\theta_i$ to a low dimensional behavior space, the QD problem is to find the highest performing solution $\theta_i$ for every value of $\mathbf{m}$. Since $\mathbf{m}$ is a continuous variable, estimating a good solution for every point in behavior space requires infinite memory and is intractable. The QD problem is usually relaxed by discretizing $\mathbf{m}$ into a finite number of cells $M$, represented as a $k$-dimensional archive $\mathcal{A}$. The optimization

problem then becomes $\mathbf{max} \sum_{i=0}^{M} f(\theta_i)$, where $\theta_i$ is a solution whose measures $\mathbf{m}(\theta_i)$ fall into cell $i$. Differentiable Quality Diversity (DQD) [9] considers the problem where the objective and measure functions are differentiable, which provides gradients $\nabla f(\cdot)$ and $\nabla \mathbf{m}(\cdot)$. Quality Diversity Reinforcement Learning (QD-RL) considers a subclass of problems that can be framed as sequential decision making tasks with exploitable Markov Decision Process (MDP) structure. Instead of optimizing for a single optimal policy, the goal is to find a collection of high performing policies that are diverse with respect to embedding functions $\mathbf{m}$ that encode behavior information in a low-dimensional space. QD-RL algorithms vary in implementation and leverage recent works in both on-policy and off-policy RL [22, 30, 29, 23]. Proximal Policy Gradient Arborescence (PPGA) [1], on which we build here, is a state of the art QD-RL method that combines on-policy reinforcement learning with DQD. It maintains a current search policy corresponding to some policy $\pi_{\theta_\mu}$ in the archive. The objective and measure gradients $f$ and $\mathbf{m}$ are estimated for this policy and used to branch off solutions into nearby cells. The information on which branched policies most improved the archive, where policies that land in new cells are favored, is used to derive a first-order gradient estimate of maximum archive improvement. On-policy RL is used to walk the search policy towards those promising new regions of the archive that have yet to be explored. PPGA has produced state of the art results on challenging locomotion tasks. A particularly nice property of this method is that the first-order approximation of the gradient w.r.t. archive improvement improves with higher archive resolution. Since training diffusion models require large datasets, upsampling the archive resolution in PPGA generally results in better performance and allows us to produce more data for the diffusion model.

**Archive Distillation**    Archive distillation is the process by which a collection of solutions is distilled into a single model. This is particularly useful in the QD-RL domain, since having a single policy with full coverage of the behavior space, the ability to interpolate between points in the behavior space, and compose different behaviors to produce new behaviors, makes the model more versatile, memory efficient, and easily deployable in downstream tasks. Prior works predict that a form of the manifold hypothesis (Elite Hypervolume), exists because policies that map to the same low-dimensional behavioral space, despite occupying different niches, may share certain traits. Follow-up works attempt to either find such low-dimensional representations or illuminate them by searching over the manifold directly [11, 24]. Contemporary work in the QD-RL domain has shown success in archive distillation on difficult RL problems. [8] jointly produces an archive using a state of the art QD-RL method Policy Gradient Assisted Map Elites [22] and distills the archive into a single behavior-conditioned model. [20] uses a variant of Map Elites to produce a collection of policies that perform well in uncertain environments, and distills these policies into a single Decision Transformer [3]. Previous methods have also applied Generative Adversarial Networks to generate a diverse collection of policies for a robot ball-throwing task [16]. Here, we aim to improve on generative models applied in the QD-RL domain by scaling the representational capacity of our model to collections of deep neural networks, while simultaneously maintaining the performance and diversity of the original archive.

**Diffusion**    Diffusion models have become state of the art in image generation. Denoising Diffusion Probabilistic Models (DDPM) are a class of generative models that iteratively denoise noise sampled from an isotropic Gaussian. The iterative denoising process is a powerful mechanism that has been shown to produce state of the art results on computer vision benchmarks [6]. Numerous methods have made improvements on DDPMs that address some of the shortcomings of these models compared to other generative methods. [6] shows that classifier-guidance can be applied at test-time to improve the quality of the generated samples. [26] showed that, by relaxing the Markov assumption in the forward diffusion process, one can significantly improve inference-time by downsampling the number of diffusion steps while maintaining most of the sample quality. Multiple refined methods for sampling from a diffusion process have been proposed, including [27], [19] and [17]. However, here we are not particularly concerned with sampling efficiency, and thus use the method proposed in [26]. [25] showed that diffusion can be performed on the latent space of a pretrained Variational Autoencoder.

**Graph Hypernetworks**    Hypernetworks are models capable of estimating the weights of a secondary network [12]. When conditioned on task identities, these can achieve continual learning by rehearsing task specific weight realizations [33]. Graph hypernetworks were originally introduced for architecture search in image classification [18] and have been shown to be trainable with RL to estimate variable architecture policies for locomotion and manipulation [14] and for quadrotor control [13].

# 3 Method

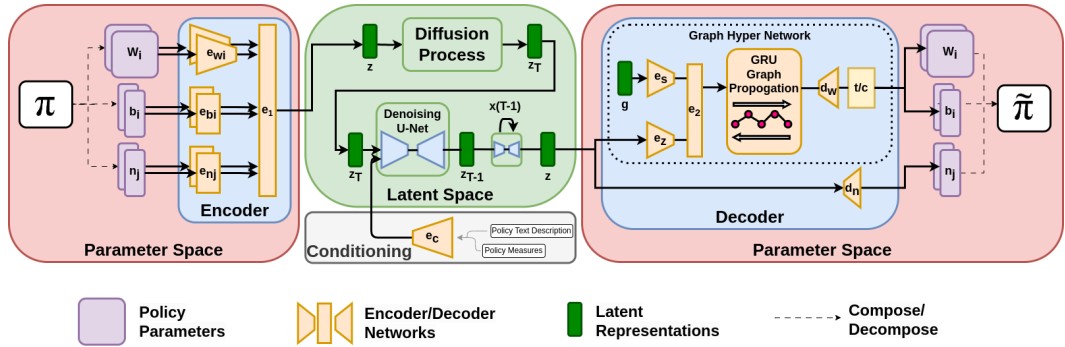

Figure 1: **Structure of our model as an encoder (left) and decoder (right).** During encoding, policies are split into layers and encoded separately. The encodings are concatenated together and fed into a final layer to produce a latent representation. The conditional diffusion model samples a latent code $z$ from the latent representation. During decoding, a graph hyper network jointly decodes the weights and bias parameters from $z$, and the policy network architecture graph $g$, while normalization parameters are directly decoded from $z$.

**Policy Compression**    Following [25], we compress the archive $\mathcal{A}$ into a lower dimensional space using a variational autoencoder (VAE). A policy consists of $l$ layers, each containing a weight matrix $W_i$ and bias vector $b_i$, $1 \leq i \leq l$. In the encoder $\mathcal{E}$, the features of each $W_i$ and $b_i$ are extracted using a convolutional neural network and a fully connected network, respectively. These features are concatenated together and fed into a final fully connected layer, that produces an intermediate latent representation $z \in \mathbb{R}^{h \times w \times c}$. The decoder $\mathcal{D}$ contains a conditional graph hypernetwork and an observation normalizer decoder, $d_n$, which takes in the latent code $z$ and produces the reconstructed policy $\pi'_i = \mathcal{D}(z_i)$. The conditional graph hypernetwork estimates the policy network's parameters while $d_n$ estimates the observation normalizing mean and variance parameters. Our conditional graph hypernetwork is based on the implementation in [14]. While the original graph hypernetwork is meant to estimate the parameters of variable architectures (represented as graphs), we freeze the input architecture graph $g$. This is set to be the architecture graph of all the networks in the archive, which in our case is represented as $\{0, 128, 128, a\}$. Here 0 indicates the input node, the following 128's represent the hidden layer nodes and $a$ is the output node and equals the action space dimension. Further, we add a latent encoder $e_z$, and with the graph encoder $e_s$, a concatenated encoding is fed to the gated graph network [34]. This mechanism lets us condition the parameter estimation on the latent $z$. Together, $\mathcal{E}$ and $\mathcal{D}$ form the VAE architecture, which optimizes the objective

$$L_{VAE} = L_{rec}(\pi'(a|s), \pi(a|s)) + D_{KL}\left[\mathcal{D}_\phi(z|\pi)||\mathcal{N}(0, I)\right] \tag{1}$$

**Policy Diffusion**    We hypothesize that the denoising process can be used to produce high quality policies given a dataset of behaviorally diverse policies parameterized by deep neural networks. We describe how the DDPM formulation can be applied to such datasets. Diffusion models consist of a forward process $q$ that iteratively applies noise $\epsilon_t$ to a sample $x_0$ from the original data distribution $x_0 \sim q(\mathbf{x})$. The noise $\epsilon_t$ at each timestep is sampled according to a variance schedule $\{\beta_t\}_{t=1}^T$

$$\epsilon_t = q(x_t|x_{t-1}) = \mathcal{N}(x_T; x_{t-1}\sqrt{1 - \beta_t}, \beta_t\mathbf{I}) \tag{2}$$

making the forward process Markovian. The reverse or generative process $p(x_T)$ reverts noise from an isotropic Gaussian into a sample $x_0$ in $q(\mathbf{x})$ and contains a similar Markov structure.

$$p_\theta(x_0) = p(x_T)\prod_{t=1}^T p_\theta(x_{t-1}|x_t), \quad p_\theta(x_{t-1}|x_t) = \mathcal{N}(x_{t-1}; \mu_\theta(x_t, t), \Sigma_\theta(x_t, t)) \tag{3}$$

Here, $x_0$ is a latent code $z$ representing some policy $\pi_\theta(a|s)$, rather than the policy parameters itself. Thus, the diffusion model instead learns to capture the distribution of the much lower dimensional $z$, analogous to the Elite Hypervolume hypothesized in [32].

[15] makes a connection between the reverse process and Langevin Dynamics, where $p_\theta(x_{t-1}|x_t)$ is the learned gradient of the data density. When $x$ represents neural network parameters, the diffusion model learns the score function of the *optimal policy distribution*, and iteratively refines the noisy policy parameters $x_t$ towards this distribution. When conditioning the policy $x$ to match a specific behavior $\mathbf{m}$ i.e. $p_\theta(x_{t-1}|x_t, \mathbf{m})$, this gradient can be thought of as the gradient of the maximum a posteriori (MAP) objective over the distribution of policies that exhibit behavior $\mathbf{m}$ with respect to policy parameters $x_t$. Thus, our diffusion formulation draws inspiration from Bayesian Methods, where $p(x_T, \mathbf{m}) \prod_{t=1}^{T} p_\theta(x_{t-1}|x_t, \mathbf{m})$ resembles an iterative gradient descent from a policy randomly initialized from an isotropic gaussian $x_T \sim \mathcal{N}(0, \mathbf{I})$ towards the mode of the posterior distribution over high-performing policies with behavior $\mathbf{m}$.

**Training Procedure**    We follow the training procedure in [25]. We first train an autoencoder according to the objective in Eq. 1. A random batch of policies and their observation normalizers $(\pi_\theta, \eta)$ are sampled from the archive and fed into the encoder $\mathcal{E}$ to produce latents $\mathbf{z} = \mathcal{E}(\pi_\theta, \eta)$. The decoder $\mathcal{D}$ then reconstructs the policies and their respective observation normalizers from the latents $(\pi'_\theta, \eta') = \mathcal{D}(\mathbf{z})$. To simplify training, on some tasks, we normalize the archive dataset by subtracting out the per-parameter mean and dividing by the per-parameter variance. This results in an autoencoder over parameter residuals relative to the original per-parameter mean and variance, which we keep for decoding. For training the latent diffusion model, we sample a batch of policies and their respective obs normalizers, measures, and text labels $(\pi_\phi, \eta, \mathbf{m}, \mathbf{y})$. The policies and measures are first encoded into latent vectors and measure embeddings respectively $\mathbf{z} = \mathcal{E}(\pi_\theta, \eta), \tau_{\psi_m}(\mathbf{m})$, where $\tau_{\psi_m}$ is a trainable encoder. These are subsequently fed into the diffusion model, where the latents are conditioned on the measure embeddings using the cross-attention mechanism. We uniformly sample $\mathbf{t}$ from $\{1, ..., T\}$ for the batch and regress the predicted noise vectors according to the latent diffusion training objective

$$L_{LDM} := \mathbb{E}_{\mathcal{E}(\pi_\phi), \epsilon \sim \mathcal{N}(0,1), t} \left[ ||\epsilon - \epsilon_\theta(z_t, t, \tau_{\psi_m}(\mathbf{m})||_2^2 \right] \tag{4}$$

In the case of language-conditioned diffusion, the measures are replaced with text labels that are encoded using a Flan-T5-Small encoder ([4]), which is fine-tuned end-to-end using the loss in Equation 4 to produce text embeddings $\epsilon_\theta(z_t, t, \tau_{\psi_y}(y))$ that condition the diffusion process.

## 4   Experiments

In our experiments, we wish to analyze our model's performance on the following: 1. archive compression while maintaining original performance, 2. measure and language conditioning to produce policies with specific behaviors, and 3. sequential behavior composition to produce new behaviors. Since PPGA was evaluated on the Brax [10] environments Humanoid, Walker2D, Halfcheetah, and Ant, we evaluate our model on the same four environments. For each environment, the reward function encourages stable forward locomotion while minimizing energy consumption, and the observation and action spaces are continuous. The measure functions are the proportion foot contact time of each leg of the robot over the entire trajectory and are thus bound to [0, 1], where 0 indicates the foot never touched the ground, and 1 indicates the foot never left the ground. Policies with different measure functions correspond to different locomotion gaits. The dimensions for the (observation, action) spaces are: Humanoid (227, 17); Walker2d (17, 6); Halfcheetah (18, 6); Ant (87, 8). Every policy in the archive has two hidden layers of 128 neurons each, followed by an output layer matching the action space dimension. While there are recent works that perform archive distillation on these tasks [8, 20], they produce a very different dataset of policies using different underlying QD-RL methods. The Quality Diversity Transformer, for example, uses evolutionary strategies with explicit optimization towards policies with low-variance behaviors, whereas PPGA uses first-order gradient approximations and makes no such explicit optimization towards low behavior variance. As any comparison of distillation methods is relative to the archive being distilled, we are unable to make any direct comparison to these methods.

**Performance and Accuracy Experiments**    We evaluate our model's ability to reconstruct the performance and behavior of policies in the archive to high precision. Following [20], we first downsample our trained archives for each task into 50 equally-spaced geometric regions using Centroidal Voronoi Tessalation - Map Elites (CVT-ME) [31]. Each region has a policy $\pi_\theta$ from

the original archive and a corresponding behavior $< m_1, ..., m_k >$ for a $k$-dimensional archive that lies in the center of that region. These policies' behaviors are used as conditions to produce 50 measure-conditioned policies $\pi_{\theta_1}, ..., \pi_{\theta_{50}}$. Each policy is then rolled out 50 times, and the objective and measure functions $f(\pi_{\theta_i}), \mathbf{m}(\pi_{\theta_i})$ are computed as the average over 50 episodes. These values are then used to compute the reward ratio, which is the average performance of the generated policies over the original ones: $r = \frac{\sum_{i=1}^{50} f(\pi'_{\theta_i})}{\sum_{i=1}^{50} f(\pi_{\theta_i})}$ .

The reward ratio alone can be misleading if all the generated policies have high performance but incorrect measures w.r.t. the measure conditions. For example, generating 50 best-performing policies with the same measures despite sampling different measure conditions would lead to a large reward ratio but no diversity. Thus, we also track the Jenson-Shannon (JS) Divergence between the distribution of measures produced by the generated policies and the original policies. We refer to this as the measure divergence. We report the JS divergence instead of the KL divergence because there is no clear preference between the KL divergence directions, and because some experiments produce policies with near-zero measure distribution overlap, for which the JS divergence is upper bounded by $ln(2)$, and the KL divergence is arbitrarily large. We perform this once every 10 epochs over the course of training and present the results in Fig. 2. Humanoid, Walker2D, and Halfcheetah achieve a reward ratio of near $\sim 1.0$ while reaching a measure divergence of $10^{-2}$. Ant achieves a reward ratio of $\sim 0.75$ with a measure divergence of $\sim 0.1$. We expect a higher measure divergence on Ant given that it is a 4-legged locomotor and thus has twice as many measures as the other environments.

**Archive Reconstruction**    At test time, we analyze the ability of the latent diffusion model to reconstruct the entire original archive. We take the measure vector $< m_1, ..., m_k >$ corresponding to cell $c_i$ and use it as a condition to produce policy $\pi'_{\theta_i} \forall c_i \in \mathcal{A}$. If the resolution of archive $\mathcal{A}$ is $d^k$, where $d$ is the discretization level and $k$ is the number of measures, this gives us a collection of $d^k$ policies. These are rolled out to compute $f$ and $\mathbf{m}$, and inserted into a new, empty archive $\mathcal{A}'$ according to the standard insertion rules in QD optimization, where only the best solution for a cell $i$ is stored when two solutions map to the same location in behavior space. The reconstructed archive will thus have some number of unique solutions such that $|\mathcal{A}'| \leq d^k$ with a QD Score of $\sum_{i=0}^{|\mathcal{A}'|} f(\pi'_{\theta_i})$. To make an informative comparison between the reconstructed and original archives, we plot their cumulative distribution functions (CDF) (Fig. 4). These not only encapsulate coverage and QD-score information, but also tell us how the policies are distributed with respect to the objective.

On all tasks, the policy distribution of the reconstructed archive closely matches that of the original one. On Halfcheetah, we are able to almost exactly reproduce the original archive. On all other tasks, we lose some density in the low performing regions of the archive, but consistently match policy density in the higher performing regions. Fig. 3 tells a strikingly similar story, in that our model first fills the central and often higher performing regions of the archive before moving on to the fringe regions to improve diversity. Both results seem to suggest that the diffusion model first learns common features corresponding to high performance across all policies, and then proceeds to learn aspects of individual policies that make them behaviorally unique. Table 1 provides a quantitative view of the CDF plots. We report the QD-scores and coverage on both the original and reconstructed archives for all tasks. Following [20], we report the Mean Error in Measure (MEM), which is the average $L_2$ distance between all generated policies' and original policies' measures in archives $\mathcal{A}'$ and $\mathcal{A}$, respectively: $\text{MEM} = \mathbb{E}\big[||\mathbf{m}(\pi_{\theta_i}) - \mathbf{m}(\pi'_{\theta_i})||_2^2\big]$.

**KL Coefficient Ablation**    A KL penalty (Eq. 1) is used to regularize the latent space. [25] used a relatively small penalty coefficient of $10^{-6}$ to prevent information loss due to overly compacting the latent space. We wish to similarly quantify the information density of our dataset and the effects of stronger latent space regularization on VAE model performance. Fig. 5 shows the VAE reward ratio and JS Divergence for larger values of the KL coefficient. Overall, our findings are in line with [25] - stronger regularization results in a loss of information, thus reducing our VAE's ability to reproduce policies from the original dataset. For all other experiments, we fix the KL coefficient to $10^{-6}$.

**GHN Size Ablation**    We examine the effect of model size on our model's ability to reproduce the archive. We chose three different values for the number of neurons in the hidden layers of the hypernetwork in the decoder and keep the diffusion model size fixed. The results are shown in Table 5 for the Humanoid environment. The QD Ratio indicates the QD score of the reconstructed archive over the original archive's QD score. The compression ratio is calculated as the number of parameters in the decoder plus the number of parameters in the diffusion model, divided by the sum

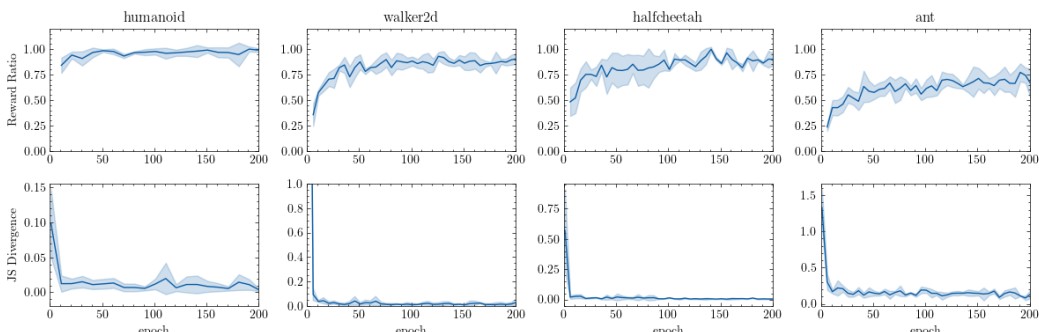

Figure 2: **Performance of our method on four benchmark environments.** Higher reward ratios correspond to better performing policies. Lower JS divergence corresponds to more precise measure reconstruction. See Section 4 for a description of the experimental method and limits of these performance metrics.

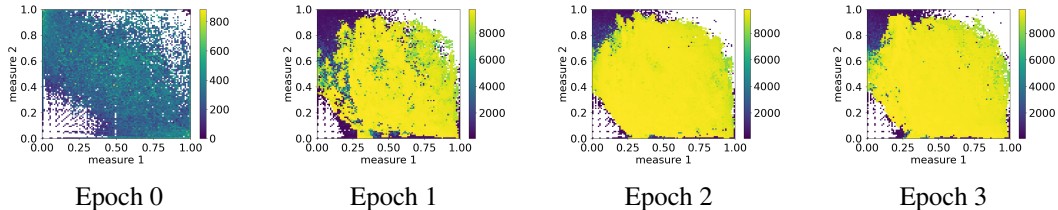

| Epoch 0 | Epoch 1 | Epoch 2 | Epoch 3 |

Figure 3: **Heatmaps of generated archives throughout training.** We visualize the reconstructed archive generated by our model over the course of training. The axes correspond to the measure values (proportion foot contact times). Each point corresponds to a policy in the archive and its color represents performance (reward) for forward locomotion. Coverage and average reward increases rapidly in early epochs to approximately match the original archive. Reward ratios and JS divergence continue to improve slowly with more epochs, as shown in Figure 2.

| Task | QD Score $(\times 10^7)$ | Coverage (%) | Compression Ratio | Reconstructed QD Score $(\times 10^7)$ | Reconstructed Coverage (%) | MEM |
|------|------|------|------|------|------|------|
| Humanoid | 8.08 | 90.79 | 13:1 | $7.4 \pm 0.11$ | $84.35 \pm 1.01$ | $0.237 \pm 0.03$ |
| Walker2D | 3.12 | 85.68 | 11:1 | $3.00 \pm 0.02$ | $83.65 \pm 0.14$ | $0.269 \pm 0.07$ |
| Halfcheetah | 11.4 | 96.94 | 14:1 | $11.1 \pm 0.01$ | $94.55 \pm 0.11$ | $0.15 \pm 0.01$ |
| Ant | 3.44 | 71.03 | 13:1 | $2.54 \pm 0.07$ | $52.66 \pm 1.46$ | $0.57 \pm 0.08$ |

Table 1: **Latent Diffusion QD Metrics.** The QD Score and Coverage columns are calculated by rolling out the policies in the original archive training dataset. The compression ratio is calculated as the number of parameters in the decoder plus the number of parameters in the diffusion model, divided by the sum of parameters of all policies in the original archive. The Reconstructed QD Score and Reconstructed Coverage are calculated by rolling out the policies that were generated by our model, conditioned on the measures from the original archive.

of parameters of all policies in the original archive. In general, we find that the MEM decreases and QD ratio increases with larger model size, at the expense of the compression ratio. Nonetheless, even the largest diffusion model (43.7 million parameters) achieves a compression ratio of 8 to 1, while reproducing 94% of the original archive with low measure error. In situations where model size is not a significant constraint, picking the largest model may be the best option, as it nearly recovers the original archive's performance and covers all relevant parts of the behavior space covered by the original dataset.

**Sequential Behavior Composition** To test our model's ability to successfully compose an arbitrary chain of generated behavior policies, we design an experiment where the episode, which naturally

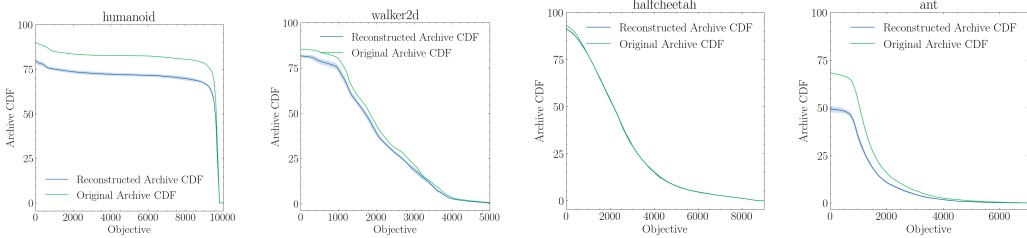

Figure 4: **CDF of policies for all tasks.** For each task, the y axis represents the percentage of the archive grid that has at least the corresponding amount of returns on rollout (shown on the x axis).

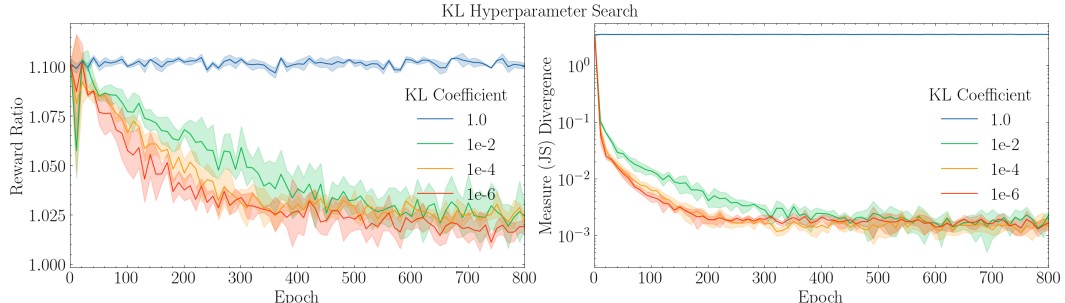

Figure 5: **Reward Ratio and JS divergence of VAE reconstructed policy for different KL coefficients.** A very large KL coefficient results in our method producing policies very close to the average policy. This results in a reward ratio above one, at the cost of complete loss of precision in measure space, and a Measure (JS) Divergence near the maximal value of $ln(2)$. Smaller KL coefficients result in more precise policy reconstruction, while maintaining a Reward Ratio near 1.

| Decoder | Parameters | QD Ratio | Compression Ratio | MEM |
|---------|-----------|----------|-------------------|-----|
| GHN8 | 18.3M | 0.77 | 19:1 | $0.267 \pm 0.091$ |
| GHN16 | 26.8M | 0.87 | 13:1 | $0.242 \pm 0.101$ |
| GHN32 | 43.7M | 0.94 | 8:1 | $0.129 \pm 0.005$ |

Table 2: **GHN size ablation on the humanoid archive.** The Compression Ratio is rounded to the nearest integer. QD Ratio and Mean Error in Measure improve with larger GHN sizes.

terminates at $T = 1000$, is divided into four time intervals of 250 timesteps each. For each time interval, we randomly sample a measure condition **m** and use our conditional diffusion model combined with the decoder to produce an agent that exhibits behavior **m**, $\pi_\phi(a|s, \mathbf{m}) = \mathcal{D}(\epsilon_\theta(z_T, T, \mathbf{m})), z_T \sim \mathcal{N}(0, 1)$. An experiment is successful if the episode terminates no earlier than $t = 800$, implying that our model has produced all four behavior policies and successfully composed at least three of them together. We consider the trajectory length for one experiment to be the average trajectory length over 50 parallel environments, and we repeat this experiment 10 times. Using measure conditioning, we found that the success rate is 48.5%, compared to the success rate of 51% when choosing the policies with the appropriate measures directly from the archive dataset, closely matching the performance of our original archive. However, a surprising result emerges when using a "pruning strategy" with our language-conditioned model, where we essentially prune out policies with text labels that suggest they are low performing. For example, when sampling sequences of text labels, we filter out those that contain the word "fall", which indicates a policy that falls over mid-trajectory. This allows us to select better policies that don't fall over on their own, which increases the success rate of sequences to 59%. Finally, we sample sequences using only text labels that contain the term "quickly," which indicates policies that move forward quickly. This raises the success rate of sequencing four different behaviors to 79%, more than double the success rate of sampling text labels uniformly. An example of sequential behavior composition with our language-conditioned model, with corresponding frames demonstrating the motions, is shown in Figure 7. This pruning strategy is capable only with language and the emergent higher success rate demonstrates the importance of language-conditioned generative

models and their capabilities for downstream tasks. This approach is similar to the use of "quality tags," used in some diffusion models.

**Compute Resources**    Each VAE and diffusion experiment was run on a SLURM cluster where each job was allocated 6 cores of a Intel(R) Xeon(R) Gold 6154 3.00GHz CPU, a NVIDIA GeForce RTX 2080 Ti GPU, and 108 GB of RAM. Each VAE experiment took about 16h; each measure conditioned diffusion model took about 3h to train while the language conditioned diffusion runs took 4h.

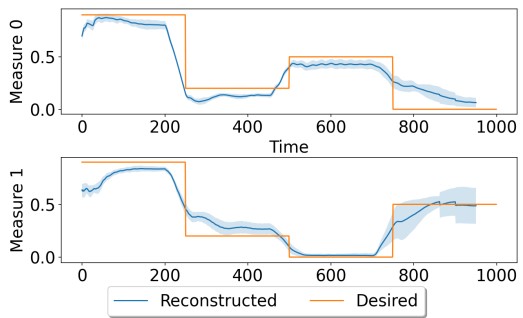
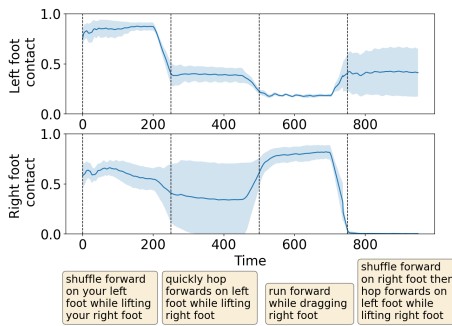

Figure 6: **Visualization of the measure values resulting from a sequence of four randomly selected desired measures (left) and text labels (right).** The measure values from a policy sequence (Section 4) are shown on the left as a function of time. These were run 10 times and the corresponding error plots are shown. The close match in measure values between the Desired values, which are used for conditioning, and the Reconstructed values show the effectiveness of our conditioning. On the right, the measure values from a policy sequence as described in Section 4 are shown as a function of time. The text labels used to produce the policy sequence are shown at the bottom of the figure. Large changes in measure values show that text conditioning is able to produce and sequence highly diverse policies. The desired behavior in both cases is changed four times throughout the episode.

## 5    Limitations

Scaling with measure dimension warrants further investigation (e.g., on ant, we see high MEM). Further experimentation with diffusion hyperparameters might produce better policy reconstructions. The language descriptions we use are currently limited and can be expanded to include a much larger corpus of descriptions. Language can occasionally under-describe the desired behavior, and therefore some descriptions can lead to undesirable generated policies. For example, Figure 6 shows higher variance in the reconstructed measures when conditioned on language. Finally, training the diffusion model requires us to first construct the policy archive by using a QD algorithm. An interesting

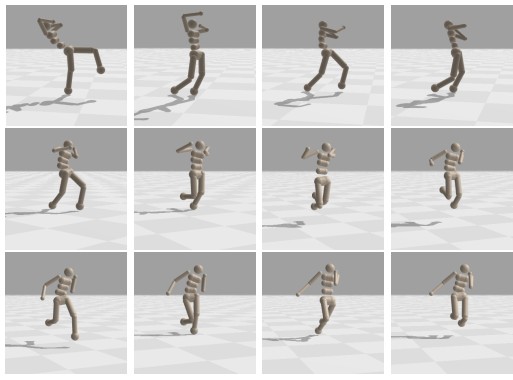
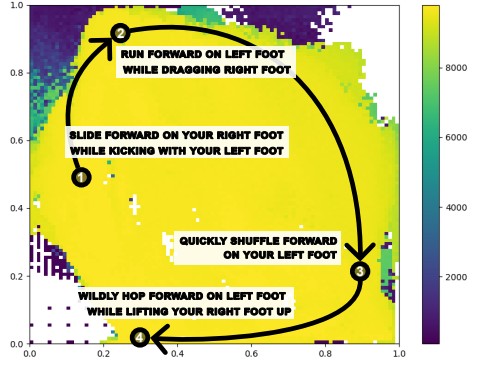

Figure 7: **Temporal behavior sequencing from text labels.** A humanoid (left) controlled by a policy sequence beginning with "slide forward on your right foot while kicking with your left foot" (top left), then "run forward on left foot while dragging right foot", then "quickly shuffle forward on your left foot", and finally "wildly hop forward on left foot while lifting your right foot up" (bottom right). Heatmap of the archive (right) showing the sequence of text labels overlaid on the measure space.

research direction would be to train this model directly, bypassing the archive construction step entirely.

# 6 Conclusion

We proposed a method that uses diffusion models to distill the archive into a single generative model over policy parameters, achieving a compression ratio of 13x while recovering 98% of the original reward and maintaining 89% of the original coverage. Our models can flexibly generate specific behaviors using measures or text, and these behaviors can be sequenced together with surprising consistency. A pleasantly surprising find was additional evidence supporting the Elite Hypervolume hypothesis proposed in [32]. From our training results and heatmap evolution over time, we see that the diffusion model first learns the general structure of what comprises a "good" policy across behavior space, and then proceeds to branch out to different areas of behavior space, implying a learning of what makes each policy behaviorally unique. As a potential future work, we can use equivariant layers such as those used in [35] instead of convolutional layers used to encode the weights and biases of policy networks. Finally, we look forward to exploring the connections this work has to other subfields, such as Meta-RL and Bayesian Learning.

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

# Appendix A   Model details and Hyper parameters

## A.1   Variational AutoEncoder

GHNx refers to a diffusion model where the decoder (the GHN) has a hidden layer with x number of neurons. The main paper shows results across tasks with a GHN16 model. We normalize the archive

| name | value |
|---|---|
| $z$ dimension | 64 |
| Encoder hidden dimension | 64 |
| Obs Normalizer encoder hidden dimension | 64 |
| KL coefficient | 1e-6 |
| Gradient clipping | True |
| Learning rate | 1e-4 |
| Training batch size | 32 |
| GHN hidden layer size | 16 |

Table 3: Hyperparameters used to train the VAE

dataset by subtracting the per parameter mean and dividing by the per-parameter standard deviation for each policy in the archive. This is equivalent to calculating a "mean-policy" with parameters $\theta_\mu$ and subtracting it from each policy in the archive. The task for the VAE reduces to learning the parameter residuals. We enable archive-normalization on Humanoid and Ant where we saw the greatest improvement in performance and training stability, and leave it disabled on Halfcheetah and Walker2d, where we observed negligent improvements.

## A.2   Latent Diffusion Model

We utilize a UNet backbone as the architecture for latent diffusion model. We use a single ResNet block to downsample the inputs into a condensed embedding space. A spatial transformer is used to perform cross-attention with condition-embeddings in the embedding space. Finally, a single ResNet block is used to upsample the embeddings to the same dimensionality as the inputs. A single ResNet block consists of two convolutional layers plus an embedding layer for sinusoidal position embeddings. The encoder and decoder used for latent diffusion remain the same as described above.

| name | value |
|---|---|
| No. of Resnet Blocks in U-Net | 1 |
| U-Net activation | SiLU |
| Transformer heads in middle part of U-Net | 4 |
| Gradient clipping | True |
| Learning rate | 1e-4 |
| Training batch size | 32 |

Table 4: Hyperparameters used to train the Latent Diffusion Model

# Appendix B   Ablation of GHN size for the Ant archive

From the below table, we see that the performance of our method scales with the GHN size for the ant environment. For larger GHN sizes, the MEM reduces while the QD ratio increases.

# Appendix C   Rollout Videos

The entire list of rollout videos for the results shown in the paper can be found at the project site.

| Decoder | Parameters | QD Ratio | Compression Ratio | Reconstructed Coverage (%) | MEM |
|---------|-----------|----------|-------------------|---------------------------|-----|
| GHN8 | 12.0M | 0.55 | 15:1 | $38.86 \pm 0.02$ | $0.582 \pm 0.023$ |
| GHN16 | 14.1M | 0.74 | 13:1 | $52.66 \pm 1.46$ | $0.571 \pm 0.081$ |
| GHN32 | 18.5M | 0.85 | 10:1 | $58.72 \pm 0.02$ | $0.436 \pm 0.084$ |

Table 5: **GHN size ablation on the ant archive.** The Compression Ratio is rounded to the nearest integer. QD Ratio and Mean Error in Measure improve with larger GHN sizes.

Measure conditioned rollouts: https://sites.google.com/view/policydiffusion/home#h.f1x0mlj6i9wz

Language conditioned rollouts: https://sites.google.com/view/policydiffusion/home#h.ddcrcw552la1

Behavior composition rollouts: https://sites.google.com/view/policydiffusion/home#h.ups33xysbz0u

## Appendix D    Heatmaps for all tasks

The following are the heatmaps for the original archive generated by PPGA and the reconstructed archive by the LDM model. Since the measure space for the Ant environment is 4 dimensional, it cannot be visualized with our current tools. Below are the plots for Halfcheetah, Walker2d and Humanoid. These heatmaps are obtained at the end of the 200th epoch of training.

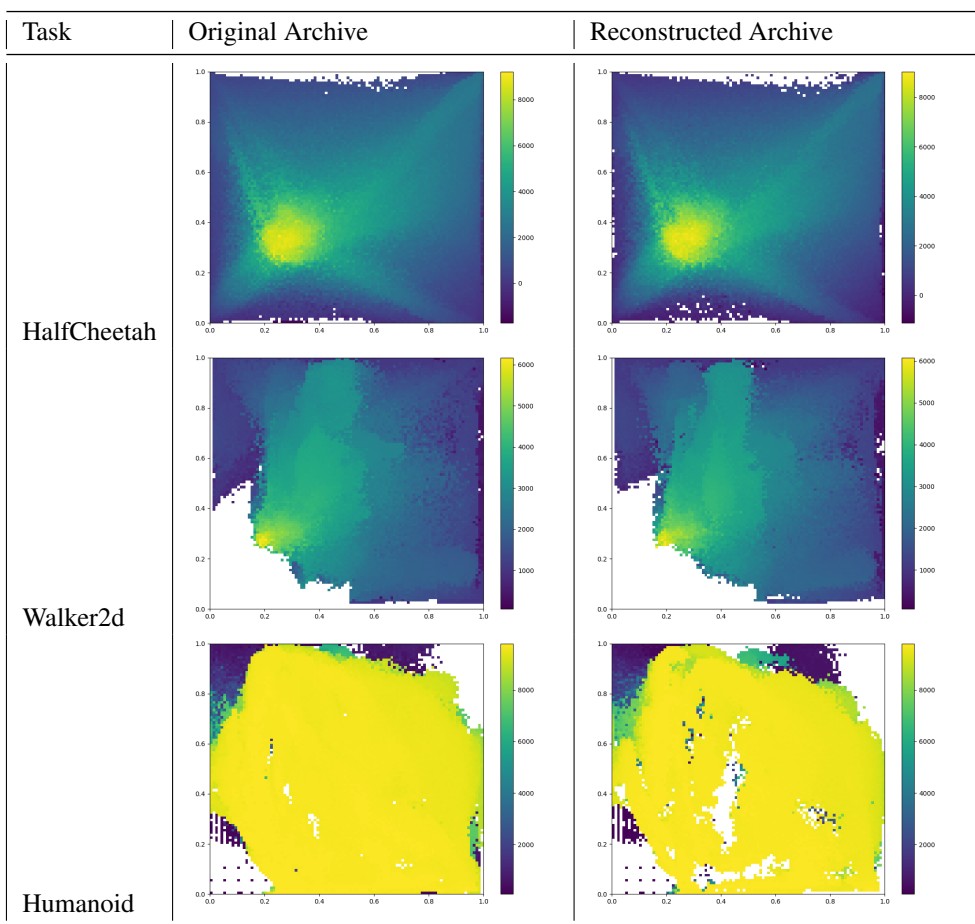

