# OpenReview forum: "Generating Behaviorally Diverse Policies with Latent Diffusion Models"
_NeurIPS.cc/2023/Conference — NeurIPS 2023 poster_

### Official Review · Reviewer_s86A · 2023-07-03

**Soundness:** 3 good
**Presentation:** 4 excellent
**Contribution:** 3 good
**Rating:** 5
**Confidence:** 3

**Summary:**

The paper presents a diffusion-based approach to compressing a policy archive discovered by a Quality-Diversity RL algorithm. The diffusion model operates in the latent space of a VAE and achieves high levels of compression together with a reasonable level of reconstruction. The algorithm is evaluated on a collection of 4 Brax environments.


**Strengths:**

- Clearly written paper and description of the approach
- Promising results showing high levels of compression on some Brax environments, together with good levels of reconstruction and coverage
- Strong visualizations of behavior during training and different synthesized behaviors

**Weaknesses:**

- The approach studied in the paper is limited to compression of the original archive. Equally, the sequential behavior composition experiments only reproduce what was possible with the original archive. An interesting next step would be understanding if the diffusion model can generalize to novel measure vectors or language instructions.
- The paper assumes very small 2-layer, 128-width MLPs trained by the QD algorithm, it is unclear if the algorithm can scale to larger and more representative networks
- High loss in diversity particularly on the ant environment, in Table 1.
- Line 8 in the Abstract, should clarify exactly what environments the authors see the compression ratio/coverage on

Minor:
- Line 42: typo in ‘uspample’
- Scale is hard to see in Figure 3, axes should also be described
- The idea of compressing policies into a single diffusion model is related to [1] which compresses offline RL datasets into a single diffusion model, [1] also achieves around 13x compression.
- It would be helpful to also indicate the level of compression in Table 1.

[1] Synthetic Experience Replay. Cong Lu, Philip J. Ball, Yee Whye Teh, Jack Parker-Holder.


**Questions:**

- The ablations on network capacity in Table 2 are only run on humanoid which works well, it would be interesting to understand if the large loss in diversity on the ant environment can be alleviated by higher network capacity or better representation learned by the VAE.
- Can the diffusion model generalize to novel conditions and improve on the original archive?
- Section 3: The convolutional layers used to encode the weights and biases of the network could be less appropriate than equivariant layers such as those used in [1].

[1] Permutation Equivariant Neural Functionals. Allan Zhou, Kaien Yang, Kaylee Burns, Yiding Jiang, Samuel Sokota, J. Zico Kolter, Chelsea Finn.


**Limitations:**

The limitations of the method are discussed well in the paper.

---

> ### Author Rebuttal · Authors · 2023-08-09
>
> **The approach studied in the paper is limited to compression of the original archive...**
>
> We agree that generalization to novel language instructions is an interesting direction for future work. However, generalizing to novel measure dimensions is not possible with the environments available. The measure functions in our policy datasets function similar to class categories in computer vision tasks i.e. it is not possible to generalize to novel measures in the same way that generative models trained on CIFAR10 cannot generalize to the CELEB-A dataset unless these datasets were combined apriori. However, similar to how vision models can generate images with different categories of objects drawn in the same image in different locations, we attempt to show with sequential behavior composition that different parts of a fixed-length trajectory can be filled in with arbitrarily different behaviors without catastrophic failure. Prior methods show the difficulty of this task and solve it by explicitly optimizing for sequential composition of different policies during training [1, 2].
>
> [1] Peng, Xue Bin, et al. "Deepmimic: Example-guided deep reinforcement learning of physics-based character skills." ACM Transactions On Graphics (TOG) 37.4 (2018): 1-14.
>
> [2] Krishna, Lokesh, and Quan Nguyen. "Learning Multimodal Bipedal Locomotion and Implicit Transitions: A Versatile Policy Approach." arXiv preprint arXiv:2303.05711 (2023).
>
> **The paper assumes very small 2-layer, 128-width MLPs ...**
> To address larger and different types of policy networks, we believe differences may arise in the encoding and decoding of these policies. For decoding, we believe that graph hypernetworks can scale well to different types of networks. It was shown in [1] that hypernetworks can estimate the weights to larger and more representative networks. [2] Further shows that these hypernetworks can estimate weights of larger MLP policies as well.  For encoding, an alternative to CNN based encoding can be using neural functional networks as described in [3]. These permutation equivariant neural functional networks seem to learn functions based on CNN networks weights very well.
>
> [1] Knyazev, Boris, Michal Drozdzal, Graham W. Taylor, and Adriana Romero Soriano. "Parameter prediction for unseen deep architectures." Advances in Neural Information Processing Systems 34 (2021): 29433-29448.
>
> [2] Hegde, Shashank, and Gaurav S. Sukhatme. "Efficiently Learning Small Policies for Locomotion and Manipulation." In 2023 IEEE International Conference on Robotics and Automation (ICRA), pp. 5909-5915. IEEE, 2023.
>
> [3] Zhou, Allan, Kaien Yang, Kaylee Burns, Yiding Jiang, Samuel Sokota, J. Zico Kolter, and Chelsea Finn. "Permutation equivariant neural functionals." arXiv preprint arXiv:2302.14040 (2023).
>
> **High loss in diversity particularly on the ant environment, in Table 1.**
> We believe that the loss in diversity on the ant environment is a result of there being a large number of poor performing policies that do not share parameters with other high performing policies in the dataset, and that this is not an issue with the proposed method itself. From the CDF plots in figure 4, we see that the diffusion model recovers most, if not all, of the higher performing policies, and loses diversity only where lower performing policies are concerned. If the proposed method struggled specifically on ant for some reason i.e. the higher dimensional measure space, we would expect an equivalent drop in diversity along all levels of policy performance. From our GHN size ablation, we see that coverage on Ant correlates strongly with model capacity. Thus, we expect to be able to recover most of the original archive’s coverage with larger models.
>
> **Line 8 in the Abstract,..**
> Thank you for pointing this out. We will clarify this point in the revised manuscript.
>
> ### Questions:
>
> **The ablations on network capacity in Table 2 are only run on humanoid...**
>
> Thank you for your suggestion. We have conducted this ablation and have added it to the attached file. Scaling up the GHN size in the decoder does indeed alleviate the performance drop of our model on the Ant environment. Further, we see that the compression ratio is still reasonable at larger GHN sizes.
>
>
> **Can the diffusion model generalize to novel conditions and improve on the original archive?**
> We believe that our model can interpolate well within the bounds of the measure space described in the global response above. Unfortunately, with the current specified measures, it is not possible to generalize outside these bounds because it is not physically possible. For example, a measure of 1.2 implies 120% foot contact time with the ground. Measure functions in QD behave as discrete categorizations of behavior, similar to how image categories function in computer vision tasks. However, an interesting research direction would be to add new measure functions online to the archive, filling it in with new policies while jointly using these new policies to train the diffusion model, in order to increase its expressivity and dimensionality of the behavior space.
>
> **Section 3: The convolutional layers used to encode the weights and biases of the network could be less appropriate than equivariant layers such as those used in [1].**
>
> Thank you for this suggestion! We were not aware of this paper prior to submission and believe this would be a promising direction to improve our method’s representation capacity in the future.
>
> ### Minor weaknesses:
>
> The corrected figure has been added to the attached additional figures page.
>
> **The idea of compressing policies into a single diffusion model is related to [1] ..**
>
> Thank you for bringing this recent work to our attention. We agree that compressing offline RL datasets and compressing policy datasets is related, so we will add a citation to this work.

---

> > ### Comment · Reviewer_s86A · 2023-08-13
> > **Thank you**
> >
> > Thank you for the responses to the review. I will maintain my current score.

---

### Official Review · Reviewer_GgS4 · 2023-07-07

**Soundness:** 2 fair
**Presentation:** 2 fair
**Contribution:** 2 fair
**Rating:** 3
**Confidence:** 4

**Summary:**

Quality-Diversity Reinforcement Learning generates a set of policies (here called the archive) that are learned to produce varied behaviors in the environment.  These archives can be large, and this paper aims to compress a previously learned archive into a single model by leveraging a conditional diffusion model.  Each of the policies in the archive is first represented in a latent space by using a variational auto-encoder that reconstructs the weights and biases of the policy's neural network.  Once the encoder and decoder of the policies have been trained, a diffusion model is then fit to the encoded latents, and conditioning information is used to help guide sampling.  The paper explores conditioning based on the measure of a policy (where the measure is a set of functions used to split the policies into different regions of behavior space), or text descriptions of the policies.  The paper shows that the learned diffusion model can generate policies that return similar rewards as the original archive in aggregate, and also have high overlap with the conditioning measure when the sampled policy is executed again in the environment.

**Strengths:**

This paper proposes an interesting application of powerful generative models to fit a Quality-Diversity archive of policies.  The ability to reconstruct the full archive from a single generative model increases the practicality of QD approaches to skill discovery in reinforcement learning.  The paper does a good job highlighting this contribution, and it is indeed an intriguing direction.

**Weaknesses:**

- Evaluation of the model is thorough, in that ablations and several domains are explored, however it is difficult to assess the quality of the approach given that no alternative approaches are attempted.  In line 210, the paper argues that other approaches to archive distillation are not comparable because the underlying archive is different.  I disagree: since the main contribution of this paper is a distillation method, it should be able to compared to other distillation methods when the archive is held fixed.
- I find the metrics used to evaluate the model difficult to interpret, perhaps related to the lack of baseline.  The Mean-Error in Measure (MEM) metric described by the paper is a reasonable one, but the paper does not describe what measures are used in the various tasks, so interpreting the scale of MEM is difficult.  QD score is similarly difficult to interpret.  Is a decrease of $0.6 \times 10^7$ QD score a meaningful one?
- The experiment on sequential behavior composition is similarly difficult to interpret.  Is 80% success of 4 consecutive behaviors good?  Perhaps including the success rate of the original archive would be a good start.
- Graphs in Figure 3 are not immediately interpretable without prior exposure to that form of QD visualization.

**Questions:**

- I find the design of the network encoder to be strange.  Why are convolutional filters applied to the weight matrix of an MLP when encoding?  Is there some spatial structure there to be captured?
- What dependence does the diffusion model have on the underlying archive to be compressed?
- Where do text descriptions of behaviors come from? Are policies labeled after learning, or are the labels pre-specified with the desired measures?  Is it easy to create this conditioning data?

**Limitations:**

The authors have addressed the method's limitations.

---

> ### Author Rebuttal · Authors · 2023-08-09
>
> **Evaluation of the model is thorough,...**
>
> There are two archive distillation methods most similar to ours, DCG-Map-Elites [1] and the QD-Transformer [2]. However, both of those methods are concurrent with ours and the code was not available for either of them at the time of submission. There are also design differences that make direct comparison difficult. The algorithm proposed in [1] jointly fills the archive and distills the policies into a descriptor-conditioned actor policy. Replacing the descriptor-conditioned actor with our diffusion model would fundamentally be a different algorithm and produce different results. We understand the reviewer’s concerns for baselines. However, to the best of our knowledge, archive distillation is a relatively novel idea and only starting to be investigated. We hope to be a baseline for future methods.
>
> [1] Faldor, M., Chalumeau, F., Flageat, M., & Cully, A. (2023). MAP-Elites with Descriptor-Conditioned Gradients and Archive Distillation into a Single Policy. arXiv preprint arXiv:2303.03832.
>
> [2] Macé, Valentin, Raphaël Boige, Felix Chalumeau, Thomas Pierrot, Guillaume Richard, and Nicolas Perrin-Gilbert. "The Quality-Diversity Transformer: Generating Behavior-Conditioned Trajectories with Decision Transformers." arXiv preprint arXiv:2303.16207 (2023).
>
> **I find the metrics used to evaluate the model difficult to interpret,..**
>
> Please see the answer in the global response at the top.
>
> **The experiment on sequential behavior composition....**
>
> We have performed sequential behavior composition evaluations on the original archive, and obtain the same results as reported in the paper by the generative model. There are a number of prior works [1,2] where the primary or secondary goal was to compose behaviors with explicit optimization on the distilled model. Therefore, we believe that 80% is an impressive success rate given that our method was not explicitly designed or optimized to compose policies sequentially.
>
> [1] Peng, Xue Bin, et al. "Deepmimic: Example-guided deep reinforcement learning of physics-based character skills." ACM Transactions On Graphics (TOG) 37.4 (2018): 1-14.
>
> [2] Krishna, Lokesh, and Quan Nguyen. "Learning Multimodal Bipedal Locomotion and Implicit Transitions: A Versatile Policy Approach." arXiv preprint arXiv:2303.05711 (2023).
>
> **Graphs in Figure 3 are not immediately interpretable...**
>
> Please see the answer in the global response at the top.
>
> ### Questions
>
> **I find the design of the network encoder to be strange...**
>
> Deconvolutional layers are typically used for parameter generation in hypernetworks [1][2]. For symmetry, we chose to use convolutional layers for encoding. Our experiments show that convolutional layers are sufficient to perform this encoding task, which indicates that useful policy encoding does not depend on the interactions of parameters that are distant from each other in the weight matrices, a type of loose spatial structure. The use of convolutional layers has the advantage of reducing the total parameter count when compared to an MLP-based encoder. The encoder is not used during policy generation, and consequently has not been a major focus in this work. In future work, encoding may be further improved by incorporating advancements from recent work in [3].
>
> [1] Knyazev, Boris, Michal Drozdzal, Graham W. Taylor, and Adriana Romero Soriano. "Parameter prediction for unseen deep architectures." Advances in Neural Information Processing Systems 34 (2021): 29433-29448.
>
> [2] Hegde, Shashank, and Gaurav S. Sukhatme. "Efficiently Learning Small Policies for Locomotion and Manipulation." In 2023 IEEE International Conference on Robotics and Automation (ICRA), pp. 5909-5915. IEEE, 2023.
>
> [3] Zhou, Allan, Kaien Yang, Kaylee Burns, Yiding Jiang, Samuel Sokota, J. Zico Kolter, and Chelsea Finn. "Permutation equivariant neural functionals." arXiv preprint arXiv:2302.14040 (2023).
>
> **What dependence does the diffusion model have on ..**
>
> We believe that in order to generalize and smoothly interpolate between behaviors as shown in the sequential behavior composition task, the diffusion model requires archives with higher resolutions, on the order of thousands of policies. The “Covariance Matrix Adaptation” line of work i.e. CMA-ME [1], CMA-MAEGA [2], etc. tend to favor larger archives, since having more cells results in higher quality gradient estimates in the evolutionary adaptation component of these algorithms. Methods such as PGA-ME [3] tend to favor lower resolution archives because they jointly optimize the entire archive and do not maintain a local search distribution that optimizes for where in the archive to explore next. We use PPGA [4] for our work, which under the hood runs CMA-MAEGA, thus favoring large archives, and has also produced SOTA results on the locomotion tasks, thus making it the ideal choice for use with diffusion models.
>
> [1] Fontaine, Matthew C., Julian Togelius, Stefanos Nikolaidis, and Amy K. Hoover. "Covariance matrix adaptation for the rapid illumination of behavior space." In Proceedings of the 2020 genetic and evolutionary computation conference, pp. 94-102. 2020.
>
> [2] Fontaine, Matthew, and Stefanos Nikolaidis. "Differentiable quality diversity." Advances in Neural Information Processing Systems 34 (2021): 10040-10052.
>
> [3] Lim, Bryan, Manon Flageat, and Antoine Cully. "Understanding the Synergies between Quality-Diversity and Deep Reinforcement Learning." arXiv preprint arXiv:2303.06164 (2023).
>
> [4] Batra, Sumeet, Bryon Tjanaka, Matthew C. Fontaine, Aleksei Petrenko, Stefanos Nikolaidis, and Gaurav Sukhatme. "Proximal Policy Gradient Arborescence for Quality Diversity Reinforcement Learning." arXiv preprint arXiv:2305.13795 (2023).
>
> **Where do text descriptions of behaviors come from?..**
>
> Please see the answer in the global response at the top.

---

> > ### Comment · Reviewer_GgS4 · 2023-08-16
> >
> > Thank you, authors, for your clarifications.  My major concerns about baselines have been addressed by the clarification that other similar work is concurrent.  Additional clarifications on the metrics and figures are also helpful.

---

### Official Review · Reviewer_mRc8 · 2023-07-15

**Soundness:** 2 fair
**Presentation:** 3 good
**Contribution:** 3 good
**Rating:** 6
**Confidence:** 3

**Summary:**

This paper tried to solve the high space complexity in Quality Diversity and proposes a method that uses diffusion model to distill the archive into a single generative model based on policy parameters.

**Strengths:**

* This paper leverages the generation power of diffusion model and condenses one model instead of thousands of policies.
* This paper is well structured.

**Weaknesses:**

* Quality Diversity is not well-known. It's better to include a background section instead of including basic knowledge in the related work.
* It's confusing no baselines can be directly compared. The metrics can not only be rewards but also space efficiency.

**Questions:**

* Need more explanation and comparison in experiment
* There are some environments for quality diversity like QD-GYM. What's the performance of the method in QD-GYM?

---

> ### Author Rebuttal · Authors · 2023-08-09
>
> **Quality Diversity is not well-known. It's better to include a background section instead of including basic knowledge in the related work.**
> We have included a brief background section in our related work section. Please see the global response for a detailed explanation of measures we used in Quality diversity. We would be happy to split into its own section and extend it where space permits.
>
>
> **It's confusing no baselines can be directly compared. The metrics can not only be rewards but also space efficiency.**
>
> Unfortunately, possible baselines are very recent work (within 3 months of the submission), and code is not available to easily reproduce them. We show space efficiency in Table 2. We would be happy to add a note to Table 1 to point to Table 2 for details on space efficiency.
>
> ### Questions:
>
> **Need more explanation and comparison in experiment**
>
> Thank you for your feedback. Please see the global response for more details. We will add additional analyses to the experiments section, space permitting, and hope this resolves any additional questions or points of confusion the reviewer may have.
>
> **There are some environments for quality diversity like QD-GYM. What's the performance of the method in QD-GYM?**
>
> We use QDax[1], which re-implements all of the tasks available in QD-Gym in a GPU-accelerated environment. If our method were to be run on QD-Gym the results should be identical, but the experiments would take significantly longer to run.
>
> [1] Lim, Bryan, Maxime Allard, Luca Grillotti, and Antoine Cully. "Accelerated Quality-Diversity through Massive Parallelism." arXiv preprint arXiv:2202.01258 (2022).

---

### Official Review · Reviewer_Usyj · 2023-07-19

**Soundness:** 3 good
**Presentation:** 3 good
**Contribution:** 3 good
**Rating:** 7
**Confidence:** 4

**Summary:**

This work presents a novel framework using latent diffusion models to distill the archive of polices into a single generative model over policy parameters. The latent diffusion model with VAE backbone compresses the high dimensional neural network (NN) parameters into a compact, making it possible to reconstruct policies parameterized by NN. Further, the conditioning mechanism of diffusion models is used to flexibly generate policies with different behaviors.

**Strengths:**

1.	The author(s) use latent diffusion model to compress the high dimensional neural network (NN) parameters into a compact, making it possible to directly generate policies parameterized by NN.
2.	The proposed framework recovers 98% of the original rewards and 89% of the original coverage while achieving a compression ratio of 13x.


**Weaknesses:**

1.	The performance and accuracy of the proposed method show significant discrepancies when applied under text conditions. Experiments show that the success rate of the method is influenced by the selection of text labels. This suggests that the model possesses limited understanding and generation capabilities in terms of language descriptions.
2.	The proposed method shows poor performance on tasks with high-dimensional measure vector, such as Ant, suggesting its limited modeling capability in high-dimensional measure spaces.


**Questions:**

1.	How many policies are used as training dataset when conducting archive reconstruction experiments?
2.	Do the dimensions of the measures used in each environment have explicit semantic information? If so, can the author(s) further demonstrate the model's generalization ability on it? (For instance, assuming that a particular dimension of the metrics represents the robot's movement speed, if the value of that dimension is set to 1.2 in the condition of the diffusion model, will the generated strategies be able to achieve speeds beyond those observed in the training dataset?) If not, how could the users employ this model to generate policies of their desired behaviors?


**Limitations:**

1.	The author(s) only demonstrated the model's ability to reconstruct the original dataset and did not conduct further experimental demonstrations regarding its generalization capability. However, the generalization ability should be an important consideration when evaluating generative models.
2.	At present, it appears that text conditioning does not show a sufficiently favorable influence within the model.

---

> ### Author Rebuttal · Authors · 2023-08-09
>
> **The performance and accuracy of the proposed method show significant discrepancies when applied under text conditions. Experiments show that the success rate of the method is influenced by the selection of text labels. This suggests that the model possesses limited understanding and generation capabilities in terms of language descriptions.**
>
> Thank you for your insights; we too found that the language-based generation capabilities are limited compared to measure-conditioned policy generation. Our hypothesis for this discrepancy lies in the fine-granularity of measure conditioning compared to the coarse-granularity of language conditioning. In order to produce large enough training data for the diffusion model, each measure is finely discretized into 100 equally spaced bins i.e. a 2D archive for a bipedal robot contains 10k cells and thus as many as 10k policies with unique measures. In practice, however, there are much fewer actual gaits than the 10,000 “unique” policies that can exist in the archive. Thus, a single language condition describing a specific high-level gait corresponds to a region of the archive, rather than a specific policy with a specific measure, as is the case with measure conditioning. This means that there is less training data for the language encoder to work with when compared with measure conditioning. Nonetheless, language conditioning is still useful since it is more intuitive than using measures, and it’s likely the case that many real world applications will not require measure-level granularity, instead opting for gait-level granularity. Improving on language-level conditioning, however, is an interesting direction of research we intend to pursue in the future.
>
>
> **The proposed method shows poor performance on tasks with high-dimensional measure vector, such as Ant, suggesting its limited modeling capability in high-dimensional measure spaces.**
>
> While we agree that the performance of ant is lacking compared to the other tasks, we do not believe that the dimensionality of ant’s measure space is affecting performance since the measure space of ant (4) is not significantly higher than the other tasks (2). In addition, the ant measures are discretized into only 10 bins as opposed to the 100 bins in the 2D archives of the other tasks, making the total maximum policy count in the datasets the same for all tasks. The CDF plots in Figure 4 show that the diffusion model recovers most, if not all, of the higher performing policies, and loses diversity where lower performing policies are concerned. If the proposed method struggled with the higher dimensionality of the measure space, we would expect an equivalent drop in diversity along all levels of policy performance. This suggests that the ant archive dataset itself contains many different policies that have parameter variations that are difficult to distill but achieve low reward.
> The MEM is also not corrected by the dimensionality of the measure space. By dividing the MEM by the measure dimensionality, we can compute a MEM per dimension of Ant that has a similar value as the other environments. Finally, an ablation on GHN size on the Ant environment and provided in the added document shows that performance can be improved with larger decoders.
>
>
> ### Questions:
>
> **How many policies are used as training dataset when conducting archive reconstruction experiments?**
> 7470 policies are used for humanoid, 9815 for half-cheetah, 8192 for walker2d, and 6180 for ant.
>
> **Do the dimensions of the measures used in each environment have explicit semantic information? If so, can the author(s) further demonstrate the model's generalization ability on it? (For instance, assuming that a particular dimension of the metrics represents the robot's movement speed, if the value of that dimension is set to 1.2 in the condition of the diffusion model, will the generated strategies be able to achieve speeds beyond those observed in the training dataset?) If not, how could the users employ this model to generate policies of their desired behaviors?**
>
> The dimensions do not have explicit semantic information. In our tasks, the measures are defined as the proportion of foot contact time for each leg of the robot over the entire trajectory. This inherently bounds the measure space to [0.0, 1.0] for each measure dimension. Consequently, there is no physical interpretation of a measure value of 1.2. The goal of each policy is the common RL objective for locomotion tasks, which is to move forward as fast as possible while minimizing energy consumption and avoiding termination conditions such as falling over. Thus, we cannot modulate the speed of the generated policies since the dataset contains policies that were trained to move as fast as possible. It would, however, be possible to modulate velocity, if forward progress was added as a measure function instead of as the objective. While it would be interesting to add forward progress as a measure function and experiment with out of distribution generalization in a separate QD paper, our focus in this work was archive distillation. Here, we achieve desired behaviors using measure or language-conditioning describing the behavior, where the behaviors in question correspond to different gaits.

---

> > ### Comment · Reviewer_Usyj · 2023-08-18
> > **Thanks detailed response**
> >
> > I truly appreciate the author's detailed response. Thanks to that, all of my questions and concerns have been thoroughly clarified.

---

### Author Rebuttal · Authors · 2023-08-09

We thank all the reviewers for their in-depth feedback. Attached here is a pdf document with additional tables and figures.

**Measures in Quality Diversity:**
Thank you for the reviewers' feedback. We see that our explanation of measures is lacking. Below is a brief explanation of what measures are used. The same edits will be made to the revised manuscript. The measure functions for all tasks in our experiments are the same – each measure function measures the proportion foot contact time for each foot on the robot. The measure functions are thus bounded to the [0, 1] interval, where 0 indicates the foot never touched the ground, and 1 indicates the foot never left the ground. We agree that QD-scores are difficult to interpret, and also have the weakness that they are sensitive to archive resolution and scale of the objective function. However, we feel it is necessary to include it in our evaluation to allow our results to be compared with prior works in QD that include the QD-score in their evaluations. A more interpretable description of our method’s performance can be found in the CDF plots in Figure 4. CDF plots show the percentage of policies that achieve an episodic reward of R or higher, for all possible R values on the x-axis. Thus, it captures aspects of both performance and diversity, as well as how the policies are distributed with respect to performance – data which is not well represented by a scalar value. We will revise the description of Table 1 to emphasize that Figure 4 provides more interpretable evaluation results.

**Visualization of the archive:**
Figure 3 visualizes an implicit archive over the course of training for the diffusion model. The axes of the figure describes the measures described above. As the diffusion model learns to better represent the policy distribution w.r.t. performance and behavior, it is capable of filling more cells (where each cell corresponds to a policy with that behavior) with high performing solutions. Near-optimal policies are represented in yellow. We appreciate the feedback and will update the caption and axes of Figure 3 to better explain this. The updated figure is in the attached additional figures pdf file.

**Text Labeling policies**
The training text labels were generated by manually labeling policies in the original archive after the archive is generated but before training the diffusion model. We labeled policies by repeatedly inspecting a rollout for the policy farthest in parameter space from all previously labeled policies. We continued this process until we had labeled 128 of the policies, and labeled the remaining policies using nearest neighbors in parameter space. Labeling took approximately 3-5 minutes per policy on average, for a rough total of 10 hours. Most of that time was spent searching for the next policy or waiting for policies to be rendered, which could be avoided with better labeling tools.

---

### Decision · Program_Chairs · 2023-09-21

**Decision:**

Accept (poster)

**Comment:**

Four experts reviewed this paper with all accepted recommendations. The area chairs agree that this work makes a very important contribution by introducing the use of diffusion models to distill the archive into a single generative model over policy parameters. The reviewers did raise some valuable concerns that should be addressed in the final camera-ready version of the paper!